# Personality Traits as Predictors of Malevolent Creative Ideation in Offenders

**DOI:** 10.3390/bs12070242

**Published:** 2022-07-21

**Authors:** Enikő Szabó, Attila Körmendi, Győző Kurucz, David Cropley, Timea Olajos, Nóra Pataky

**Affiliations:** 1Oradea Penitentiary, Parcul Traian 3, 410033 Oradea, Romania; 2Department of Personality and Clinical Psychology, Faculty of Humanities, Institute of Psychology, University of Debrecen, 4002 Debrecen, Hungary; ati.kormendi@gmail.com (A.K.); olajos.timea@arts.unideb.hu (T.O.); 3Department of Work and Organization Psychology, Faculty of Humanities, Institute of Psychology, University of Debrecen, 4002 Debrecen, Hungary; kurucz.gyozo@gmail.com; 4UniSA STEM, University of South Australia, Adelaide, SA 5001, Australia; david.cropley@unisa.edu.au; 5Department of Pedagogical Psychology, Faculty of Humanities, Institute of Psychology, University of Debrecen, 4002 Debrecen, Hungary; nora.pataky@freemail.hu

**Keywords:** malevolent creativity, malevolent ideation, personality, dark triad, offenders

## Abstract

Malevolent creativity, which can be defined as creativity that is deliberately planned to damage others, is a concept that explains how the capacity to generate novel and effective outcomes (creativity) may, on occasion, be misapplied. The present study used 130 male inmates of the Oradea Maximum Security Penitentiary in Romania to explore the ability of a set of personality variables (the dark triad, self-efficacy, and self-esteem) to predict malevolent creative ideation. The findings indicate that Machiavellianism and self-efficacy were significant predictors of malevolent creative ideation in the form of lying, while only Machiavellianism was a significant predictor of malevolent creative ideation in the form of hurting people. In addition, the present study found significant differences among subgroups in the sample, with more experienced offenders showing higher levels of malevolent creative ideation.

## 1. Introduction

The *dark side* of creativity has grown in importance as a focal point for creativity research, in particular, since Cropley, Kaufman, and Cropley [1] defined *malevolent creativity*, specifically, as “… creativity that is deliberately planned to damage others.” (p. 106). For more than a decade, scholars have explored many facets of malevolent creativity, including personality (for a discussion of malevolent creativity research from 2008, see [2]). It is notable, however, that almost all studies of malevolent creativity have examined the variables of interest in *normal*—i.e., non-criminal—samples. Thus, while there is a growing body of knowledge about the relationship between personality, cognition, and malevolent creativity among non-criminal individuals who are engaging, at best, in *hypothetical* deviant behavior (e.g., [3]), far less is known about malevolent creativity in the very individuals who may be most likely to engage in malevolent creative behavior: namely, criminals.

The main objective of the present study was to investigate the role of personality—specifically, the dark triad, along with self-esteem and self-efficacy—as a predictor of malevolent creative behavior in a criminal population. In conjunction with this objective, the study also explored the role of *criminal experience* in malevolent creativity, speculating that a longer and more violent *criminal career* would be associated with higher levels of malevolent creativity. The following sections review the current research on malevolent creativity, criminal behavior, and personality, before setting out the details of a study designed to examine personality traits as predictors of malevolent creative ideation in a sample of convicted criminals.

### 1.1. Malevolent Creativity

According to Sternberg and Lubart [4], creativity is the ability to create useful and unique things. Rhodes [5] defined creativity in terms of the four Ps: product, personal, process, and place. Cropley and Cropley [6] offered an extension of this model by adding two more components and dividing the personality component into three subcategories: personal motivation, personal attributes, and personal feelings. Guilford frequently emphasized the significance of the evaluation procedure for developing innovative ideas [7]. Another description of creativity could be conceived of as a product that stands out for its value and originality [8]. Although there is no universally acknowledged definition of creativity, it is widely understood to be a beneficial force in the workplace, in education, in cultural environments, and in technological innovation. Cognitive processes linked to creativity include cognitive flexibility [9], problem-solving [10], intelligence [11,12,13,14], divergent thinking [15], attention [16,17,18], memory [19,20,21], and imagery [22].

Despite research focusing on the positive aspects of creativity [23,24], little was known about its harmful outcomes [25,26]. McLaren was the first to mention the concept of dark creativity [27]. Recent research has revealed a growing interest in the malevolent aspects of creativity, such as unethical behavior [28], deception [29], and manipulation [29]. Cropley [1] defined malevolent creativity as the development of a product with the intent of providing novelty to the detriment of other people. Runco [30] posits that it is not the process of creativity that should be viewed as good or bad, positive or negative, but rather the motivation of the individual in releasing the product. Novitz [31] also disputes the idea that malevolent creativity is even a form of creativity. Workplaces, terrorism, and crime are all places where malevolent creativity can be found [32]. Ligon, Sporer, and Derrick [33] examine terrorist inventiveness in a focused manner. They maintain that research into creativity requires a more particular and profound understanding of the mechanism that lies behind all violent extremist organizations. By revealing how it works, future studies will be able to better manage and possibly mitigate the harm that malevolent creativity causes to others. Antagonism, hostility, and sympathy [34], terrorism [35], and possibly even videogames [36] were discovered to have a positive link with malevolent creativity. Antagonism and anger at the state may explain the variance of malevolent creativity performance [37]. Malevolent creativity levels may be influenced by the way in which individuals perceive the legality and destructiveness of their actions [38]. The goal type and the goal achievement are predictors for malevolent creativity beyond individual differences, in accordance with social informational theory [39]. Malevolent creativity, in this context, could be demonstrated by all individuals, not in the form of the “evil genius” but when informational cues exist to adopt some kinds of innovative strategies. In previous studies, malevolent creativity has been linked to emotional intelligence and approach motivation [40], with individuals showing higher levels of malevolent creativity performance in terms of fluency and originality [41]. Malevolent creativity has also been linked to the traits of physical aggressiveness [34], implicit aggression [40], and actual aggression [42], while aggression itself has been found to act as a moderator between approach motivation and malevolent creativity [41,42]. Moral disengagement mediates the negative association between malevolent creativity and authenticity. Abusive supervision shows a positive relationship with the generation of workplace malevolent creativity; individuals may try, via different means, to encourage their colleagues to behave in an undesirable manner [43]. However, relevant moderators for this close relationship were the light triad (i.e., Kantianism, humanism, and faith in humanity). In terms of divergent thinking tasks, such as the alternative uses task [44,45,46], the evidence reveals a substantial association between malevolent creativity and sex (maleness), which could be attributed to evolutionary or physiological (hormonal) causes. The fact that malevolent creativity can take a multitude of forms, not simply those scored by one of the only current assessment measures, the Malevolent Creativity Behavior Scale (MCBS) [47], such as sabotage, deception, revenge, and so on, presents a psychometric challenge. When seeking to identify reappraisals of anger-eliciting circumstances, the participants created more ideas within a malevolent creativity task and also demonstrated increased fluency of thought. When it came to the volitional reassessment of anger-eliciting circumstances, a higher degree of harmfulness of ideas that were generated with the intent of harming another person was associated with the less fluent development of ideas [37]. Reflection is needed on the relationship between dishonesty and creativity because there is a positive link between them. Dishonesty and divergent thinking have in common the fact that the thinker will break the rules in order to make new opportunities [48,49]. Liu and Ye [50] suggest that creativity can influence the motivation to behave in a dishonest way, as individuals may use it to justify their unfair behavior. Creative ideas may generate rule-breaking behaviors, which are unacceptable in terms of social norms, this having a consequent conservative effect on individuals [51]. Lying is also a technique that is widely used by malevolent creative people. People tend to lie when they are in an enriched environment [52]. The environment also acts as a mediator between creativity and cheating [53], suggesting another effect of “thinking outside the box”, within which context individuals can be free. One of the most recent studies on the subject [54] found that authenticity is negatively associated with malevolent creativity because it is closely related to morality; openness to experience has a negative relationship with malevolent creativity, indicating a negative link to morality; risk-taking is negatively associated with malevolent creativity because it can involve actions that lead to rule violations.

Taking this evidence and the reported research findings, along with the ongoing uncertainty about malevolent creativity theory and its measurement, we proposed and designed a study to identify the personality traits that are predictors of malevolent creativity. Although, at the level of theory, many articles have mentioned the existence of malevolent creativity in crime and terrorism, most of the published research has examined general populations, with only a few studies analyzing inmate populations and malevolent creativity [55].

### 1.2. Criminal Behaviour

Every act that purposefully violates the established rules and laws is considered criminal conduct. “It means that we must study all the possible data that can be causes of crime—the man’s heredity, the man’s physical and moral make-up, his emotional temperament, the surroundings of his youth, his present home, and other conditions—all the influencing circumstances …” (Lombroso, 1911, p. vii). According to biological theory, the first research in the field may be dated directly to Lombroso [56], who was the first to investigate the potential link between criminal psychopathology and physical anomalies, including skull size and facial bone abnormalities. Sheldon [57] attempted to explain delinquent behavior using human body categorizations (arguing that criminals are more muscular and their body shapes are squarer). One influential psychological theory, presented by Freud (1939–1956), explains crime as the failure of the superego, which is founded on morality and the ethical code and is being driven by the id, or innate drive. Another theory that explains criminal behavior is the social learning theory [58], which supports empirical findings [59] that delinquent behaviors are the result of the learning process, which includes the imitation of criminal behavior. Rewarding this behavior will increase its frequency while punishing it will decrease it. The criminal justice system is working hard to prevent the crimes and behaviors associated with breaking the law; nevertheless, despite various theories and programs (cognitive-behavioral, etc.), no conclusive and meaningful reductions have yet been observed. When it comes to analyzing criminal behavior, one topic that has yet to be studied is the age–crime curve. The age–crime curve has been a long-discussed topic in criminology since Quételet [60]. He was the first to notice a unimodal and positive link between age and arrest, through observation. Many sources of official criminal data—chiefly, cautions and convictions—have frequently proved that there is a substantial rise in criminal conduct throughout early adolescence, until the mid-twenties (about the age of criminal responsibility), demonstrating the concept of an age–crime curve, peaking around the mid- to late teenage years, then decreasing, steeply at first (until the mid-twenties) and then more slowly thereafter, demonstrating the notion of an age-crime curve with a peak ([61], pp. 192–195). Because variances reflect both individual developmental changes and historical developments over time, explaining the age–crime curve is difficult. A debate arose on the matter as a result of the absence of conclusive data. Since the early 1980s, Gottfredson and Hirshi [62] have stated that examining an individual’s frequency of offending is futile since crime as a function of age follows a “unimodal” curve, the form of which is invariant over time and space. According to Sampson and Laub [63], the incidence of offending varies significantly for some forms of crime and certain categories of offenders, owing to the wide range of individual criminal conduct that may be evinced across a lifetime. According to Francis et al. [64], focusing on the totality or frequency of offending is incorrect since it supports the notion that crime patterns (or convictions) are unchanging and constant. Because there are so many and varied types of crimes and criminal behavior patterns, a typology approach to understanding crime and its causes is far more useful. Terrie Moffitt [65], who classified the two types of offenders as life-course persistent offenders and teenage limited offenders, is one of the most well-known authors in the field of offender taxonomy in recent years. Le Blanc [66] confirms that the age–crime curves of the two generations of adjudicated delinquents have uniform shapes throughout time [67]: “… peak rates would arise from variations in the intensity of offending by a fairly fixed group of active offenders, with individuals’ frequency rates increasing during the juvenile years and then gradually declining with age” ([68], vol. 1, pp. 23–24). Leung [69] conducted the first systematic economic investigation and found that because a high fraction of criminals has previously been arrested, the age–crime profile will therefore decline within a specific age range, due to the selection effect: fewer offenders are arrested in this age range, which indicates that the age–crime profile can be generated merely by fluctuations in the intensity rate. Similar to the way that the age–crime curve can elucidate an individual’s criminal path, recidivism is a concept closely linked to this path, based on the recurrence of criminal acts committed by a person over their lifetime [70]. Langan and Levin [71] demonstrated that nearly 60% of those who were freed were reincarcerated within three years. This is a global issue since the incidence of recidivism is a major topic in the criminal justice system; there is still no clear proof of how to successfully minimize the incidence of reoffending. The role of demographic characteristics, also known as static factors, such as age, ethnicity, race, and others in predicting recidivism has been documented in the literature [72]. Multiple recent meta-analyses [73,74] have discovered that these static characteristics, including a young age, race, and being male, are linked to violent reoffending. Additional data suggest that age, in particular, is a strong predictor of recidivism in the case of inmates with mental illness [75].

Eisenman has provided a fairly realistic description of how malevolent creativity operates, using 9 study cases [76]. As he demonstrated in his paper, malevolent creativity is a very subtle form of aggression that is directed toward others in order to exact retribution or sabotage someone else. Later research has shown a strong association between malevolent creativity and the trait of physical aggressiveness [34], implicit aggression [40], and actual aggression [42]; it has also been established that aggression has a moderating influence on the link between approach motivation and malevolent creativity [41,42]

Finally, because malevolent creativity implies harm and, therefore, violence, and because many of the inmates in this study committed violent crimes, there is value in examining the relationship between malevolent creative ideation and violence.

**Hypothesis** **1.** 
*Malevolent creative ideation will be highest in individuals who were first convicted in adolescence (age group: 16–20).*


**Hypothesis** **2.** 
*Individuals with more convictions will show higher levels of malevolent creative ideation than individuals with fewer convictions.*


**Hypothesis** **3.** 
*Violent offenders will show higher levels of malevolent creative ideation than non-violent offenders.*


### 1.3. The Dark Triad

The dark triad is a personality trait axis that incorporates narcissism, psychopathy, and Machiavellianism. These characteristics all have one thing in common: they flout social expectations. The core nucleus consists of callous manipulation [77]. There is a great deal of evidence in the literature suggesting that these are three distinct but overlapping qualities. Based on their observations, the constellation of the dark triad was first identified by Paulhus and Williams [78]. They observed several startling similarities between the three constructs: “Despite their diverse origins, the personalities composing this Dark Triad share a number of features. To varying degrees, all three entail a socially malevolent character with behavior tendencies toward self-promotion, emotional coldness, duplicity, and aggressiveness” ([78], p. 557). Previously, genetic evidence of dark triad features was reported [79,80,81], exclusively for the callous core [82,83] in monozygotic and dizygotic twins, to contain a heritable component, validating the univariate behavioral genetic model. These are significant results supporting the nature vs. nurture argument regarding the dark triad’s beginnings, bolstering the idea of an inherited character. Various associations between childhood aggression and intimate partner violence have been noted [84,85]. Over almost two decades, there has been a dispute about the dark triad’s overlapping nature; according to O’Boyle et al. [86], Machiavellianism and narcissism have a moderate link, while psychopathy and the other two qualities have a significant correlation. Vize et al. [87] conducted a meta-analysis and discovered that narcissism is distinct from Machiavellianism and psychopathy, and that there was less overlap with narcissism when the common variance of Machiavellianism and psychopathy was controlled. The interrelationships between narcissism, Machiavellianism, and psychopathy are supported by other studies, published later [88]. A lack of empathy is also a common trait among the dark triad of characteristics. According to Black, Woodworth, and Porter [89], these three traits show a similar failure to perceive emotional vulnerability in others, as well as a perception of others as being weak. There was a considerable difference in the dark triad between genders. Men outperformed women in all three dark triad traits [90]. There was no association between intelligence and the dark triad traits [91], implying that the concept of the “evil genius” has no real foundation; all three traits were attributed to negative effects on work behavior, particularly in terms of counterproductive behavior [92]; emotional intelligence had no relationship with narcissism but a negative relationship with the other two traits of Machiavellianism and psychopathy. There is no evidence to support the theory of the dark side of emotional intelligence, neither as an ability nor as a characteristic, as there is in the case of intelligence [93]. However, when it comes to the relationship between dark triad traits and misconduct or immoral actions, there is a growing body of research exploring its connection. Although all three traits in their characteristic definitions are intercorrelated with antisocial behavior, they are functioning within parameters where, for various reasons, they have no problem with not being law-abiding. Azizli et al. [94] showed that there are strong and positive relationships between antisocial inclinations, as reflected by the dark triad traits, and behavioral factors such as misconduct and a proclivity for high-stakes deceit. Other authors [95,96] tested an evolutionary framework considering the dark triad qualities as adaptive in terms of a self-serving, exploitative lifestyle.

### 1.4. Machiavellianism

Cristie and Geis [97] gave birth to the concept of Machiavellianism as a personality trait. Machiavellianism is named after the famous Renaissance politician and influential statesman, Niccolò Machiavelli. He wrote his magnum opus, The Prince, after having been exiled from Florence. The book presents strategies and tactics for a successful reign. Machiavellianism, therefore, is associated with cynicism, a lack of morality, and a strong sense of tactics. Machiavellianism is also defined by the absence of empathy, emotional detachment, and a willingness to exploit others as primary characteristics [97,98]. In addition, Machiavellianism is also acknowledged as employing strategies to influence people for personal benefit without considering others, such as strategic planning [99]. The development of Machiavellian interpersonal strategies and a cynical view of human nature were substantially linked to a bad home environment, loneliness, and parental neglect. These are the repercussions of a lack of parental love and affection shown in not meeting a baby’s or toddler’s basic requirements. Furthermore, punishment has aided the development of deceptive and exploitative interpersonal strategies. Because the offender learns that their wrongdoings cannot be forgiven, they will continue to experience dysfunctional negative feelings, and they will learn that only irrational behavior will allow them to achieve their objectives [100]. There were also positive connections observed between aggression and aggression-related qualities [95,101,102], violence [103,104], risk-taking behavior [105,106], bullying, and delinquent conduct [101,107,108]. Machiavellianism is also defined as a lack of sincerity and a disregard for ethical considerations [96]. It has been related to a variety of diverse forms of impulsivity [83]. There is a correlation between aggressive verbal expression, physical aggressiveness, and the use of automobiles to demonstrate aggression [109]. Cognitive empathy has a close relationship with Machiavellianism, indicating that those who can predict and fully describe the actions of others are thereby capable of manipulating them. Following other studies on the genetic subdomains, researchers concluded that of the three constructs, only Machiavellianism can be altered as a result of adverse experiences [79,83]. In relation to creativity, Machiavellianism shows no link with negative creativity [90,110], nor with positive creativity or creative achievement [111]. Although empirical research suggests that persons who are strongly Machiavellian have weaker fluency and show less originality in divergent thinking tasks [112] than people who score lower in terms of Machiavellian tendencies but they also give more answers that indicate malicious actions directed at others [110].

**Hypothesis** **4.** 
*Machiavellianism will significantly predict all forms of malevolent creative ideation.*


### 1.5. Narcissism

The desire to be admired and an excessive sense of self-importance characterize narcissism. Narcissus, a hunter in Greek mythology, gives his name to the trait. He was described by the writer Conon, a contemporary of Ovid, as a lover of beauty who thought he was the most attractive of them all and expected others to commit suicide in his honor. There are two types of narcissism: grandiose narcissism and vulnerable narcissism. High degrees of narcissism are associated with aggressive driving and verbally expressed aggression [109] and violence [113]; it is also correlated with functional impulsivity [83]. Exploring the relationship between creativity and narcissism, we may argue that the two have a reciprocal relationship. On the one hand, creative accomplishments provide a favorable impression of ability and counteract narcissism [114]. These two constructs have produced a wide range of outcomes in the literature, from finding no link [111] to finding a significant association [115]. Despite the fact that narcissists frequently score highly on self-reported creativity tests [114,116] and positive creativity [110], these findings are inconsistent when compared to objective measurement [117]. The strongest evidence was discovered between narcissism and creativity, according to all three dark triad personality traits, and was moderated by multiple factors, such as the creative domain and self-reported or performance-reported measures, according to a recently published meta-analysis. These findings highlighted narcissistic characteristics, particularly grandiosity, as well as a tendency to exaggerate one’s own capabilities and abilities, as being positive variables for creativity [118]. It is possible that having the highest scores regarding vocally hostile expressions while driving is linked to egocentrism and the narcissist’s need to protect their bodily integrity [109]. Because narcissists have low self-esteem and utilize defensive techniques to compensate for their vulnerabilities, the theory of threatened egoism could provide an explanation as to why they act violently [119].

**Hypothesis** **5.** 
*Narcissism will significantly predict all forms of malevolent creative ideation.*


### 1.6. Psychopathy

Psychopathy is a widely studied concept in terms of personality, in social and forensic settings. It was described for the first time by Cleckley [120] as a personality disorder in certain case studies, highlighting its malevolent character. It is also important to mention that psychopathy is included in the Diagnostic and Statistical Manual of Mental Disorders, Fifth Edition, in Section 3, with the alternative models of personality disorders [121]. Psychopathy was also found to have a strong link with overall sexual desires, as well as a wider spectrum of sexual fantasies, including sadomasochistic, impersonal, and adventurous themes. These findings are also consistent with gender disparities; however, men also have the same kinds of sexual fantasy themes [96]. Psychopathy was associated with dysfunctional impulsivity [83]. Deviant sexual fantasies, racing, cheating, schadenfreude, financial misconduct, and cyberbullying were all found to be predictors of psychopathy [88]. Negative creativity is also associated with psychopathy [110,122]. Psychopaths are unlikely to participate in mental, imaginative, or divergent thinking processes that are typical of creativity, due to their characteristics. We can conclude that they will demonstrate low creative achievement, their interest being focused on practical activities [123]. Major evidence suggests that psychopathy is directed toward someone, with the intent of doing harm [112], or it has a negative relationship with creativity [111,117]. The meta-analysis also shows an association between psychopathy and self-reported creativity, in the context of creative activities and creative achievement [118]. The findings are also consistent with the uniqueness of subclinical psychopathy, implying that psychopaths are aware of their strengths and skills, without the need for approval from others.

**Hypothesis** **6.** 
*Psychopathy will significantly predict all forms of malevolent creative ideation.*


### 1.7. Other Personality Constructs

#### 1.7.1. Self-Esteem

Self-esteem [124] is the attitude that we have toward ourselves, which can be a state or a trait. The first can be evaluated in three separate domains, such as performance, social context, and appearance, whereas the trait of self-esteem is measured as a global structure with no distinct subgroup. Self-esteem was shown to be positively influenced by maternal sensitivity in a strong, direct way in a child population, suggesting that it is of developmental importance in early childhood. However, the results also showed that self-esteem enhanced verbal creativity in children in a direct way [125]. High levels of self-esteem were correlated with the dark triad of personality traits, suggesting that having a highly positive attitude about the self may help an exploiter to persevere in the face of social rejection and revenge [95]. The intrinsic motivation hypothesis of creativity was put forth by Amabile [126]; it claims that while being intrinsically motivated is advantageous to creativity, being externally motivated is not necessarily a bad thing. External motivators may occasionally help internal motivation, which in turn encourages innovative activity. Extrinsic motivators, for instance, can encourage creativity if they increase the creator’s enthusiasm for the endeavor, offer insightful knowledge, and encourage autonomy.

It is, therefore, plausible that self-esteem would boost creativity because people then believe that they are capable of solving problems in novel ways and have confidence in expressing novel ideas and methods. High levels of self-esteem are positively related to creativity [127] because it can help to adopt strategies to approach goals, not to avoid them [128]. Low self-esteem is an obstacle to effective treatment participation because intervention necessitates intrinsic motivation, as well as the offender’s involvement in correcting their behavior and thoughts [129]. Again, low self-esteem may be correlated to reoffending in the case of sexual abuse because guilt and shame are associated with it [130,131]. A recent meta-analysis summarized the last 25 years of research into the relationship between delinquency/crime and self-esteem, indicating a negative but small significant association; the effect size increased when it was only associated with delinquency [132]. High levels of self-esteem help individuals to persist in the task they are involved in, rather than encouraging them to give up, as is the case in those with low self-esteem [133]. Self-esteem, along with cognitive flexibility, is also found to be a significant mediator between creativity and openness to new experiences [134]. The data suggest that by incorporating creative theatre, poetry therapy, and music therapy methodologies, a self-esteem development program can effectively raise self-esteem in a college student population [135].

**Hypothesis** **7.** 
*Self-esteem will significantly predict all forms of malevolent creative ideation.*


#### 1.7.2. Self-Efficacy

The view and conviction that someone possesses the appropriate skills and can mobilize them effectively to achieve a specific action are referred to as self-efficacy. This socio-cognitive theory was created by Bandura [136], who suggested: “that self-efficacy allows a better understanding and analysis of individuals’ behavior”. The self-efficacy research domain addresses, in general, school performance, prosocial problem-solving, and career activity. Bandura thought [137] that “in conventional careers, the belief in their own efficacy … influences the course of action they pursue, the level of determination”. In laboratory settings, self-efficacy had a strong relationship with performance. However, no significant result was found between creative self-efficacy and general self-efficacy in the student population [138]. Monetary gain from crime, through the ability to evade apprehension and punishment, and the display of criminal skills or expertise can increase self-efficacy [139]. The way that juvenile delinquents perceive how to control their antisocial behavior would be related to the goals that they set in the future. Those juvenile delinquents with higher prosocial self-efficacy had higher prosocial aspirations for the future, spent less time in a residential placement, and had lower rates of recidivism [140]. Those who were incarcerated and had low levels of self-efficacy were more likely to commit rule violations, face new criminal charges and return to prison than those with higher levels of self-efficacy, who indicate compliance with mandated supervisory obligations [141]. More problem-focused copers have higher levels of self-efficacy, suggesting the possession of skills “to deal with” issues [142]. Those who reported traumas also could adopt higher levels of self-efficacy, in order to cope with trauma-associated situations [142,143]. Additionally, career self-efficacy can be increased in juvenile offenders, as Allen and Bradley [144] showed after a 12-week career-counseling intervention. Research on criminal populations has evidenced that they view themselves as being successful at crime, with high levels of criminal self-efficacy, despite the problems that they have with the law. In their interviews, inmates claimed that they could learn from their previous mistakes so that, in the future, this would give them the opportunity to refine their tactics and procedures, to become more efficacious. This is in accordance with Badura’s [137] socio-cognitive theory that “failure can, paradoxically, raise efficacy through the belief that better strategies will bring future success”. In terms of offender population self-efficacy, in relation to conventional tasks and pursuit, this was associated with desistance [145,146].

**Hypothesis** **8.** 
*Self-efficacy will significantly predict all forms of malevolent creative ideation.*


## 2. Materials and Methods

### 2.1. Procedure

The Babeş-Bolyai University of Cluj-Napoca’s Scientific Council granted this research ethical approval (registration number 4420/20 March 2018). The research was conducted from 14 October 2019 until 28 February 2020. The questionnaires were filled out using the paper and pencil method. The cohort was recruited through an announcement calling for prisoners to participate; therefore, all those prisoners who expressed a willingness to be included in the research were admitted. They agreed to all the inclusion and exclusion criteria and signed a consent form. According to the internal system procedure, they were each given three credits for participation.

The IBM SPSS statistics software version 22 was used for the statistical analysis. The statistical analysis was conducted in two ways, both descriptive and inferential, as described in Section 3 below.

### 2.2. Participants

There were initially 181 convicts included in the research, but only 130 completed our survey instruments fully. Participants were male inmates at the Oradea Maximum Security Penitentiary who had been charged with a variety of offenses, ranging from minor theft to major crimes or multiple crimes. The participants ranged in age from 21 to 66 years old, with an average age of *M* = 37.2 years (*SD* = 9.95). Of those within the sample, 16.9% had only completed primary school, 30.8 percent had completed high school, 13.8 percent had graduated from college, and 8.5 percent had a higher degree. Table 1 contains the sample’s additional demographic characteristics.

### 2.3. Measures

Malevolent creative ideation was assessed using a Romanian translation of the Malevolent Creativity Behavior Scale (MCBS; [47]). This 13-item scale evaluates behaviors on three sub-scales: hurting others (Cronbach’s α = 0.80), lying (Cronbach’s α = 0.76), and playing tricks (Cronbach’s α = 0.61). The first subscale (hurting others) consists of six items (e.g., “How often do you think of new ways to punish people?”), the second subscale (lying) consists of four items (e.g., “How often do you tell lies without fear of being nailed?”), and the third subscale (playing tricks) consists of three items (e.g., “How often do you play tricks on people as revenge?”). Participants responded on a 5-point, Likert-type scale (0 = never; 4 = usually).

The demographic questionnaire, a complex questionnaire for gathering information about gender, age, educational level, marital status, criminal history, substance abuse, etc., was composed by the authors to evaluate general data regarding the inmates.

The “Dirty Dozen” [95] contains 12 items and measures three subclinical personality traits: Machiavellianism (Cronbach’s α = 0.66), psychopathy (Cronbach’s α = 0.50), and narcissism (Cronbach’s α = 0.80). The Cronbach’s alpha coefficients indicated that the Dirty Dozen had satisfactory reliability; acceptance for this is lower on the psychopathy subscale but offers acceptable reliability. Answers are given on a 7-step Likert scale, from strong disagreement to strong agreement. A Romanian translation was provided by Dragos Iliescu. The other authors, as well as Florin et al. [147], translated this into Romanian. The total DT (Cronbach’s α = 0.85), narcissism (Cronbach’s α = 0.86), psychopathy (Cronbach’s α = 0.64), and Machiavellianism (Cronbach’s α = 0.81) presented these values in a sample of 168 students.

The self-esteem scale [124] contains 10 items listed on a 4-point Likert scale, where 1 represents “I totally disagree” and 4 means “I totally agree” (e.g., “I am able to do things as well as most other people”). Internal consistency for the present study was good (Cronbach’s α = 0.79). The Romanian version of the scale, created from David’s [148] validation of the scale, was developed using the sample and included 245 subjects; the value of Cronbach’s alpha coefficient was 0.79. The scores, therefore, demonstrate good internal consistency.

**The self-efficacy scale** [149] contains 10 items (e.g., “I can always manage to solve difficult problems if I try hard enough”), which can be rated from 1 to 4, where 1 means “completely untrue” and 4 means “perfectly true” (Cronbach’s α = 0.84). The Romanian translated form was provided by David [148] who validated the measure on a sample of 234 subjects. The value of Cronbach’s alpha coefficient was 0.84; this score, therefore, indicates good internal consistency.

## 3. Results

Preliminary assessments of skewness and kurtosis indicated that the data were normally distributed and are, therefore, suitable for parametric analysis. Table 2 shows the basic descriptive data for the sample used in this study.

Moderate positive associations were identified between psychopathy, narcissism, and Machiavellianism, which finding is consistent with a prior study [78].

To test Hypothesis 1, a one-way between-groups analysis of variance was conducted to explore the impact of an offender’s age when first convicted on their levels of malevolent creative ideation, as measured by the subscales of the malevolent creativity behavior scale (MCBS). Participants were divided into three groups, according to the age at which they were first convicted of a crime (Group 1: <16 years; Group 2: 16–20 years; Group 3: >20 years). There was a statistically significant difference at the *p* < 0.05 level in scores for hurting people in the three age groups: *F*(2, 126) = 6.11, *p* = 0.003. The difference in mean scores between the groups was medium–large, with an effect size, calculated using eta squared, of 0.09. Post-hoc comparisons using the Tukey–Kramer test indicated that the mean score for Group 2 (*M* = 6.14, *SD* = 4.08) was significantly different from that of Group 3 (*M* = 3.21, *SD* = 3.73). The effect size (Cohen’s d) for the difference between Group 2 and Group 3 was 0.75 (medium–large). Group 1 (*M* = 3.73, *SD* = 4.50) did not differ significantly from either Group 2 or Group 3; however, the effect sizes (Cohen’s d) were 0.12 (Group 1–Group 2: medium) and 0.12 (Group 1–Group 2: small). Hypothesis 1 is therefore supported in relation to the MCBS subscale on hurting people. There was no noticeable relationship between the inmates’ age at first conviction and the other two subscales regarding lying and playing tricks.

To test Hypothesis 2, a one-way between-groups analysis of variance was conducted to explore the impact of the number of convictions of an offender on their levels of malevolent creative ideation, as measured by the subscales of the malevolent creativity behavior scale (MCBS). Participants were divided into four groups, according to their number of convictions (Group 1 = 1 conviction; Group 2 = 2 or 3 convictions; Group 3 = 3 or 4 convictions; Group 4 = > 5 convictions). There was a statistically significant difference at the *p* < 0.05 level in the scores for hurting people in the four groups (*F*(3, 126) = 6.84, *p* < 0.001). The difference in mean scores between the groups was large, with an effect size, calculated using eta squared, of 0.14. Post hoc comparisons using the Tukey–Kramer test indicated that the mean score for Group 1 (*M* = 2.19, *SD* = 3.33) was significantly different from that of Group 2 (*M* = 4.82, *SD* = 4.01) and Group 3 (*M* = 6.75, *SD* = 4.92). The effect sizes (Cohen’s d) for the differences between Group 1 and Groups 2/3 were 0.71 and 1.08, respectively (medium–large and large). Group 4 (*M* = 4.00, *SD* = 3.61) did not differ significantly from Group 1, Group 2, or Group 3 (Cohen’s d for these differences were 0.52, 0.21, and 0.64, respectively, ranging from small to medium). Hypothesis 2 is, therefore, supported in relation to the MCBS subscale regarding hurting people.

To test Hypothesis 3, Student’s *t*-test was conducted to compare the malevolent creativity behavior scale (MCBS) scores for non-violent and violent offenders. There was a significant difference in scores on the subscale for hurting people between non-violent (*M* = 4.65, *SD* = 4.36) and violent offenders (*M* = 3.40; *SD* = 3.78; *t*(128) = 1.75, *p* = 0.04, one-tailed test). The magnitude of the differences in the mean scores (mean difference = 1.26, 95% *CI*: −0.17 to 2.68) was small (eta squared = 0.027). Similarly, there was a significant difference in scores on the playing tricks sub-scale for non-violent (*M* = 4.19, *SD* = 2.33) and violent offenders (*M* = 3.46; *SD* = 2.25; *t*(128) = 1.79, *p* = 0.04, one-tailed test). The magnitude of the differences in the means (mean difference = 0.73, 95% *CI*: −0.08 to 1.54) was small (eta squared = 0.025). Therefore, Hypothesis 3 was not supported.

To test Hypotheses 4–8, standard multiple regressions were run to predict the three forms of malevolent creative ideation (hurting people, lying, and playing tricks) from the dark triad subscales (Machiavellianism, psychopathy, and narcissism), self-esteem, and self-efficacy. The preliminary analyses indicated no violations of the assumptions of normality, linearity, multicollinearity, and homoscedasticity.

The first multiple regression model significantly predicted malevolent creative ideation (hurting people), at *F*(5, 124) = 10.04, *p* < 0.001, *R^2^* = 0.29. Only Machiavellianism added statistically significantly to the prediction (*p* < 0.001). Regression coefficients and standard errors are shown in Table 3.

The second multiple regression model statistically significantly predicted malevolent creative ideation (lying), at *F*(5, 124) = 5.63, *p* < 0.001, *R^2^* = 0.18. Machiavellianism added statistically significantly to the prediction (*p* < 0.001), along with self-efficacy (*p* = 0.043). Regression coefficients and standard errors are shown in Table 4.

The third multiple regression model statistically failed to significantly predict malevolent creative ideation (playing tricks), *F*(5, 124) = 1.64, *p* = 0.16, *R^2^* = 0.06. Regression coefficients and standard errors are shown in Table 5.

The results of the standard multiple regressions indicate that Hypotheses 4–8 were partially supported. Only Machiavellianism significantly predicted malevolent ideation for the sub-scale of hurting people, while both Machiavellianism and self-efficacy significantly predicted malevolent ideation for the subscale of lying.

## 4. Discussion

The dark side of creativity and its predictors are becoming increasingly popular in the literature. Most studies, however, focus only on non-criminal samples. The purpose of this study was to see if classifying dark-triad personalities can predict three specific malevolent creativity ideational behaviors—lying, playing tricks, and hurting people—among a sample of convicted criminals. Self-efficacy and self-esteem were also hypothesized to be predictors of malevolent creativity.

Some of the variables (self-esteem and self-efficacy) have not been investigated before this in connection with malevolent creative ideation; therefore, our current findings stand alone and require more clarification in the context of imprisonment, which is discussed below. A recent study conducted by Jia, Wang & Lin [150] reports an association between MCBS subscale scores for hurting people and Machiavellianism (r = 0.58), psychopathy (r = 0.45), and narcissism (r = 0.38); for the lying subscale and Machiavellianism (r = 0.57), psychopathy (r = 0.43) and narcissism (r = 0.47); and finally for Machiavellianism (r = 0.51), psychopathy (r = 0.42) and narcissism (r = 0.39). However, as far as we are aware, no study has been undertaken to assess the predictive power of malevolent creative ideation regarding dark triad personality traits. Our results support the theory that those who commit crimes at an early age (16–20 years) have higher malevolent creativity ideation than older prisoners on the hurting people subscale, which partially supports our first hypothesis. A possible explanation for the differences between inadequate behavior in adulthood, in comparison with behavior during the elementary school and adolescence periods, could be due to lower age-group rule-breakers being emotionally unstable and showing high neuroticism as children; thus, they have difficulties in controlling their affective responses. Because of this instability effect, they are likely to commit age-related rule-breaking exceedingly early. Emotional dysregulation, as reported by teachers, can be a good predictor of juvenile arrest [151]; many adolescents may adopt a deficient coping strategy, such as confrontation [152] since they are incapable of dealing with negative emotions. Malevolent creativity, on the other hand, has been shown to have little relationship with emotional instability [38] but has a stronger association with psychopathy and emotional coldness. It is not surprising that older rulebreakers are associated with malevolent creativity (due to emotional stability). Another explanation comes from the shape of the age–crime curve, which is strikingly similar across the data sources, depicting the overall prevalence of criminal conduct. There is a great deal of variance in offending patterns at the individual level; also, there has not been any agreement as to what elements determine the association between age and crime. According to the theory of criminology, there is a substantial link between age and crime. Researchers have frequently seen an age–crime curve in which criminal activity rises throughout the offender’s youth, peaks in their late teens, and then swiftly declines [153,154,155]. These findings have implications for the early prevention of juvenile delinquency, as age is a significant predictor of recidivism [73,74,75]. Efforts should be focused on prevention programs to help delinquent youths to manage the dynamic factors implicated in recidivism.

Referring to the number of offenses and the relationship between higher scores on the hurting people and lying subscales, it is suggested that the higher the number of infractions committed over time, the more experience and ingenuity that one gains, not only as the result of a learning process but also as a result of perfecting ways and strategies for committing crimes.

Contrary to our third hypothesis (H3 posited that *violent* offenders would exhibit higher malevolent creativity ideation than non-violent offenders), our results indicated that those who committed non-violent crimes in fact had higher scores on the hurting people subscale. Perhaps one explanation for this result can be taken from the neurophysiology of violent offenders, which includes inhibition response impairments compared to non-violent offenders [156,157]. Prefrontal network dysfunction appears to be most specifically associated with a recurrent, impulsive subtype of aggression that may contribute to some violent behavior [158], although there is no predicting power for crime. In a meta-analysis [159], antisocial conduct was also linked to structural and functional changes in the right orbitofrontal cortex, left dorsolateral prefrontal cortex, and right anterior cingulate cortex. According to the research included in the meta-analysis, antisocial people have severe structural and functional abnormalities in their prefrontal cortex, as determined by brain imaging [158]. Those with prefrontal network dysfunction may exhibit aggressive and violent behavior, as well as an inability to self-regulate, leading to impulsive behavior. Perhaps this is the reason why violent criminals showed lower malevolent creativity ideation, due to their desire to put their plans into action immediately.

In accordance with our results, the dark triad has also previously been demonstrated to be predictive of criminal behavior. This suggests that dark personality qualities are linked to offenses that involve a victim, whether directly or indirectly [159]. Although, in our study, only Machiavellianism, of the three dark triad traits, predicted behaviors intended to hurt others in novel ways, along with lying, partially supporting our fourth hypothesis. Although it is well known, according to the research by DePaulo et al. [160], that everybody tells lies daily, the number of lies is not normally evenly distributed [161,162]. The findings of this study show that persons who are solely focused on their own ambitions and interests, who are good at tactics, who prioritize money and power over relationships, who exploit and manipulate others to get ahead (i.e., those high in Machiavellianism), are most likely to engage in hostile creative activities, such as hurting others [103] and lying [163,164]. The main aspiration for Machiavellians is to fulfill their goals, regardless of their situation (either in a sexual setting or in daily life), even if this is performed in a selfish way, and they are willing to lie to achieve their ends [164,165,166]. Th eir selfish behavior may be due to the fact that Machiavellianism has been linked with lower levels of the traits of emotional intelligence [167] and empathy [97], implying that offenders are unable to grasp others’ feelings and to put themselves in another’s position.

Narcissism had no predictive power for the malevolent behavior ideation subscales, which does not support our fifth hypothesis. Perhaps their characteristics, such as aggressiveness [113] and functional impulsivity, serve as a barrier to the realization of efficient, malevolently creative ideas [82], in comparison to those high in Machiavellianism who are good at manipulation and cunning tactics. However, inconsistent results were discovered in terms of creativity [111,115], implying that such personalities exaggerate their own creativity [118].

Contrary to what we hypothesized, psychopathy had no predictive value for malevolent creative ideation. This could be because psychopathy is linked to impulsivity [82], meaning that they have poor self-control and are unable to manage their reactions to irritation or negative emotions, and so act instinctively. Another reason is that because of certain personality traits [123], such as emotional coldness, callousness, and a lack of empathy, and because of poor conduct control and impulsiveness [78], they are unlikely to engage in the mental, imaginative, or divergent thinking processes that are typical of creativity [123]. Therefore, as in the case of narcissism, we may speculate that persons who are high in Machiavellianism have learned good tactics and can regulate themselves, which is a critical quality for being successfully malevolent.

Our seventh hypothesis was not supported; self-esteem could not predict malevolent creative ideation, perhaps because low self-esteem is associated with delinquency [168,169]. It is important to be both original and fluent in order to develop malevolent creative ideas; however, self-esteem was found to be connected with higher levels of fluency but not with originality when performing a divergent thinking task [170].

We hypothesized that self-efficacy would predict lying, hurting people, or playing tricks, but the data only supported this hypothesis for lying. These findings are consistent with the literature, implying that if one believes that one may benefit from a crime, expertise can boost self-efficacy [139]. Additionally, lying may be a protective element, but it does not carry the same weight as breaking regulations and committing new offenses [141]. Due to its importance as a protective factor in the face of criminal conduct, there is also evidence to suggest that youths with greater levels of self-efficacy are more likely to avoid participation or to cease involvement in criminal and other antisocial activities [145,171]. In accordance with these results, Walters’ [172] two investigations provide early support for the theory that inadequate self-efficacy for preventing potential police encounters plays a role in crime continuity.

In our contribution to the literature, we investigated malevolent creativity in an exclusively incarcerated population, a sample cohort that has been researched before only partially [55]. These findings are significant in terms of forensic settings and inmate phenomena and imply that recidivist offenders display their creative potential in criminal activities and that these creative behaviors are self-expression manifestations for inmates with longer criminal records.

This study may be limited by the low to moderate reliability of several subscales, including playing tricks (MCBS) and psychopathy (Dirty Dozen). Neither scale has revealed any noteworthy associations, which may be due to low reliability, which implies substantial measurement error. Other translations of these scales suggest that there are no major issues with their psychometric properties when translated from English; however, we acknowledge that future research should examine this factor more closely, especially with a larger sample.

Future research should include individuals from other prisons, to allow a more comprehensive examination of variables. Furthermore, it will be an important subject to see how gender variations in malevolent creativity behavior manifest between males and females when it comes to playing tricks, hurting others, or lying. Another limitation can be identified in the measure of malevolent creativity (MCBS), highlighting its weaknesses, focusing on only three behaviors (lying, playing tricks, and hurting others), and failing to account for other malicious creative behaviors, such as sabotage, deception, revenge, etc. These are behaviors that only partly focus on the novelty of hurting others, as Reiter–Palmon [173] notes in her statement regarding the malevolent creativity measure. We also have to mention that our results may be biased by social desirability, because the self-reported questionnaires allow for the possibility of deception, as noted by Paulhus [174], who believes that social desirability has two dimensions: impression management and self-deception. Furthermore, this tendency can be found also in subclinical personality traits, such as Machiavellianism and psychopathy, with individuals who are possibly keen to maintain a positive image of themselves; thus, they are manipulative and have antisocial goals in terms of their interpersonal interactions. Social desirability was also found to be a suppressor of reactive aggression [175]. We did not consider it to be a significant variable in terms of control since we ensured the anonymity of every participant in the study. While acknowledging that there are ongoing efforts to balance the MCBS’s reliability by using open-ended situations, the malevolent creativity test, a recently established and widely used measure, is already employed in our ongoing investigations to address the weaknesses and limitations that we observed as a result of the absence of a divergent thinking measure in this study [37]. The behavioral components of creativity should be investigated in future research.

## Figures and Tables

**Table 1 behavsci-12-00242-t001:** Criminal conviction data for the sample.

Variable	Category	*n*	%
Age at First Conviction	<16 Years (Group 1)	15	11.6
	16 to 20 Years (Group 2)	29	22.5
	>20 Years (Group 3)	85	65.9
Number of Convictions	1 Conviction (Group 1)	47	36.2
	2 or 3 Convictions (Group 2)	34	26.2
	4 or 5 Convictions (Group 3)	16	12.3
	>5 Convictions (Group 4)	33	25.4
Type of Crime	Non-Violent	52	40.0
	Violent	78	60.0

**Table 2 behavsci-12-00242-t002:** Descriptive statistics and bivariate correlations for the study variables.

Variables	Min	Max	*M*	*SD*	1	2	3	4	5	6	7
1. Hurting People	0.0	16.0	3.90	4.05							
2. Lying	0.0	16.0	5.08	3.26	0.35 **						
3. Playing Tricks	0.0	11.0	3.75	2.30	0.19 *	0.32 **					
4. Machiavellianism	4.0	28.0	11.09	6.56	0.51 **	0.38 *	0.18 *				
5. Psychopathy	4.0	26.0	11.69	5.57	0.37 **	0.24 **	0.15	0.47 **			
6. Narcissism	4.0	28.0	15.18	6.83	0.23 **	0.15	0.05	0.39 **	0.36 **		
7. Self-Esteem	14.0	37.0	26.36	3.97	−0.09	0.07	0.13	−0.03	−0.04	0.10	
8. Self-Efficacy	15.0	40.0	30.28	5.00	−0.08	0.18 *	0.06	0.02	−0.11	−0.01	0.30 **

*M* = mean; *SD* = Standard Deviation; ** = *p* < 0.01; * = *p* < 0.05.

**Table 3 behavsci-12-00242-t003:** Standard multiple regression—malevolent creative ideation (hurting people.

Malevolent Creative Ideation (Hurting People)	*B*	95% CI for *B*		*SE B*	*β*	*R* ^2^	Δ*R*^2^
		*LL*	*UL*				
Model						0.29	0.26 ***
Constant	2.37	−2.79	7.53	2.61			
Machiavellianism	0.26 ***	0.16	0.38	0.06	0.43		
Psychopathy	0.11	−0.01	0.24	0.07	0.16		
Narcissism	0.01	−0.09	0.11	0.05	0.01		
Self-Esteem	−0.05	−0.21	0.11	0.08	−0.05		
Self-Efficacy	−0.05	−0.18	0.08	0.07	−0.06		

Note. Model = “Enter” method in SPSS; *B* = unstandardized regression coefficient; *CI* = confidence interval; *LL* = lower limit; *UL* = upper limit; *SE B* = standard error of the coefficient; *β* = standardized coefficient; *R*^2^ = coefficient of determination; Δ*R*^2^ = adjusted *R*^2^. * *p* < 0.05, ** *p* < 0.01, *** *p* < 0.001.

**Table 4 behavsci-12-00242-t004:** Standard multiple regression—malevolent creative ideation (lying).

Malevolent Creative Ideation (Lying)	*B*	95% CI for *B*		*SE B*	*β*	*R* ^2^	Δ*R*^2^
		*LL*	*UL*				
Model						0.18	0.15 ***
Constant	−1.56	−6.00	2.87	2.24			
Machiavellianism	0.17 ***	0.07	0.26	0.05	0.34		
Psychopathy	0.07	−0.05	0.17	0.06	0.11		
Narcissism	−0.01	−0.10	0.07	0.04	−0.02		
Self-Esteem	0.03	−0.11	0.17	0.07	0.04		
Self-Efficacy	0.11 *	0.00	0.26	0.06	0.18		

Note. Model = “Enter” method in SPSS; *B* = unstandardized regression coefficient; *CI* = confidence interval; *LL* = lower limit; *UL* = upper limit; *SE B* = standard error of the coefficient; *β* = standardized coefficient; *R*^2^ = coefficient of determination; Δ*R*^2^ = adjusted *R*^2^. * *p* < 0.05, ** *p* < 0.01, *** *p* < 0.001.

**Table 5 behavsci-12-00242-t005:** Standard multiple regression—malevolent creative ideation (playing tricks).

Malevolent Creative Ideation (Playing Tricks)	*B*	95% CI for *B*		*SE B*	*β*	*R* ^2^	Δ*R*^2^
		*LL*	*UL*				
Model						0.06	0.03
Constant	0.39	−2.98	3.75	1.70			
Machiavellianism	0.06	−0.02	0.13	0.04	0.16		
Psychopathy	0.04	−0.04	0.13	0.04	0.10		
Narcissism	−0.02	−0.09	0.05	0.03	−0.06		
Self-Esteem	0.08	−0.03	0.19	0.05	0.14		
Self-Efficacy	0.02	−0.07	0.10	0.04	0.03		

Note. Model = “Enter” method in SPSS; *B* = unstandardized regression coefficient; *CI* = confidence interval; *LL* = lower limit; *UL* = upper limit; *SE B* = standard error of the coefficient; *β* = standardized coefficient; *R*^2^ = coefficient of determination; Δ*R*^2^ = adjusted *R*^2^. * *p* < 0.05, ** *p* < 0.01, *** *p* < 0.001.

## Data Availability

Not applicable.

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
