# Peer review of "Personality Traits as Predictors of Malevolent Creative Ideation in Offenders"

_behavsci, 2022, doi:10.3390/bs12070242_

Round 1

Reviewer 1 Report

Notes

line 100 replace "may" with "may be"

line 118 replace "existing" with "existence"

The section on criminal behaviour feels too general. If you need to remove words then this section could be cut in half

The Chronbach's alpha on the psychopathy component of the dirty dozen is very low. Given the small number of items I doubt that removing them would help. You do need to recognise the limitations though?

It may be useful to restate the hypotheses in the Results section as there are large number of hypotheses?

Is it possible to include t-tests and probabilities in Tables 3a-3c?

Comments

Overall, I found this to be an interesting read. The Introduction is a little long and not always directly linked to the research; see notes. The sample was excellent, well-done for getting inside a prison to do this research!

The Chronbach's issue needs to be at least acknowledged. I know the link between the number of items and the alpha is problematic but the reader needs to be aware of the issue.

The results were generally good though I would like to have seen t-tests and actual probability levels in the Tables I mentioned.

I am not overly-fond of correlational studies that solely employ self-report measures and it would have been interesting for the researchers to discuss possible methods for overcoming the 'social desirability' issues they acknowledged in the Discussion section.

Perhaps, they could acknowledge the limitation and consider how they would overcome this in future research. Possibly, look at future Observation Studies or they could consider including the number of convictions in the regression model. Alternatively, they may consider using performance measures of divergent thinking in the future? Or, even develop a performance measure of malevolent creativity!

These are suggestions from someone who is not familiar with the 'dark triad' so feel free to ignore!

Reviewer 2 Report

Brief summary

The present study examined predictors of malevolent creative ideation in a sample of prisoners. The list of predictors included previous criminal records, Dark Triad traits, and self-beliefs. The malevolent creative ideation was operationalized through three inter-related facets: hurting others, lying, and playing tricks. Results showed that prisoners who committed crimes at earlier ages demonstrated a higher inclination toward hurting others. In addition, Machiavellianism was a statistically significant predictor of such facets of malevolent creativity as hurting others and lying. Lastly, self-efficacy predicted such facet of malevolent creativity as lying, whereas self-esteem was not related to any facets of malevolent creativity.

Broad comments

The creative flame is so flashy and fertile that one can be easily blinded by its inexhaustible abundance in terms of educational gains, innovations, and self-actualization while ignoring its dormant potential for causing harm, destruction, and suffering. The latter domain of so-called malevolent creativity has been largely neglected for decades. However, the situation changed dramatically within the last decade, and the given manuscript aims to further contribute to the rapidly developing literature on malevolent creativity. The manuscript’s literature review is thorough and, in most cases, properly describes relevant psychological constructs and empirical evidence. The research questions addressed in the study are clearly communicated, even though minor adjustments in the justification of some hypotheses are still desired (see the next section for details). Moreover, the target sample of prisoners—persons who had an official account of offenses—is also highly relevant for the external validity of the study. The results corroborated some previous findings obtained on the general population and offer insights into how various personality variables are intertwined with separate facets of malevolent creative ideation. Nevertheless, I have some questions about the measures, results, and their interpretation that are outlined in the next section. I sincerely hope that some of them will be of value during the revision. All references below are provided primarily to unwrap and substantiate my point of view and should not be considered as suggestions for their addition to the manuscript unless stated otherwise.

Specific comments

Introduction

Q1. Line 46. Creativity can be defined as “novel ideas” (4).” As far as I know, Guilford did not claim that creativity boils down to the novelty of ideas and often stressed the importance of the evaluation process during creative ideation (e.g., Guilford, 1956).

Q2. Lines 57–58. Despite research focusing on the positive aspects of creativity (23; 24), little was known about its harmful outcomes (25; 26).” Based on the narration and cited sources, I guess the parentheses with sources 23–24 and 25–26 should be swapped.

Q3. Lines 87–88. However, a relevant moderator for this close relationship were the light personality traits.” Do “light personality traits” refer to the Light Triad (i.e., Kantianism, Humanism, and Faith in Humanity)? Unfortunately, it is not clear from the context.

Q4. Lines 88–89. On the Alternative Uses Task (44; 45) and the divergent thinking task (46) ...” The AUT is an example of a divergent thinking test. Thus, I cannot fully understand why the authors separated one from another.

Q5. Lines 91–93. The fact that malevolent creativity can take a multitude of forms, not simply those scored by the MCBS, such as sabotage, deception, revenge, and so on, presents a psychometric challenge.” The mention of MCBS comes out of the blue. Perhaps it would be better to add some general information about MCBS and its subscales before outlining empirical findings obtained with its application?

Q6. Line 96. “…to the ability to find alternative reappraisals through qualitative factors.” What was meant by qualitative factors? Unfortunately, it is not clear from the context.

Q7. Line 101. “…offering situational the sentiment of freedom.” I could not grasp the meaning of this phrase and would like to encourage the authors to articulate it more clearly.

Q8. Lines 105–110. In these lines, I could not figure out the meaning of the following phrases: (1) “so that is unaccepted by social norms, having conservative effects on individuals” and (2) “suggesting indeed the effect upon ‘thinking outside the box’, in which context they can be free”. I would appreciate it if the authors could reword or restructure the sentences to illuminate the meaning.

Q9. Lines 112–113. “…openness to experience has a neutral relationship with malevolent creativity, indicating a neutral tendency by morality.” Am I right that “neutral relationship” stands for non-significant correlation? If so, I recommend rewording it accordingly to avoid readers’ confusion.

Q10. Lines 122–192. In the section “Criminal Behaviour”, the authors analyzed various theoretical approaches to explaining criminal behavior, described findings related to the age–crime curve, and designated the problem of recidivism. After that, they formulated three hypotheses (“Hypotheses 1–3”). I have no questions concerning the accuracy of the literature summary, but I think some hypotheses require better substantiation in light of the presented evidence. The substantiation of the first hypothesis is clear since it is based on the age–crime curve extensively discussed in the preceding paragraph. The substantiation of the second hypothesis is much or less clear because it is related to the problem of recidivism discussed at the end of the section.  However, the third hypothesis seems less connected to the recapped research findings. Could the authors elaborate on why they expected that violent offenders would show higher levels of malevolent creative ideation compared to non-violent offenders?

Q11. Line 196. Correlations may appear...” What is meant here? Does it refer to the correlations among Dark Triad traits?

Q12. Lines 314–316. Meta-analysis show association between self-reported creativity, sustained on creative actives and by creative achievement (120).” I wonder if the authors could clarify what was meant in the sentence because its relation to the meta-analysis of Lebuda et al. (2021) is not straightforward.

Q13. Lines 331–333. Amabile (131) proposed the Intrinsic Motivation Hypothesis of Creativity, which states that ‘the intrinsically motivated state is conducive to creativity, whereas the extrinsically motivated state is detrimental’.” External motivation is not always detrimental to creativity. In some cases, external motivators can support internal motivation which in turn facilitates creative behavior. For example, extrinsic motivators can benefit creativity if they reinforce the creator’s interest in the process, provide useful information, and support the creator’s autonomy (Amabile, 1993).

Q14. Lines 348–350. “…self-efficacy had a strong relationship with performance and also with behavior too.” I don’t exclude I could miss something, but I don’t understand the difference between “performance” and “behavior” here. I would appreciate it if the authors could shed some light on it in a way that they see as appropriate.

Materials and Methods

Q15. Line 397. “…statistical analysis was done in two ways: descriptive and inferential.” I recommend specifying more explicitly which inferential statistics were used (e.g., ANOVA, multiple linear regression, etc.). However, I do not see it as a crucial issue and thus prefer leaving it to the authors as to whether to consider it or not.

Q16. After close examination of the measures applied in the study, I couldn’t track any articles reporting results of their adaptation and validation on the Romanian population. All references are provided for the original versions in the English language. Am I correct that almost all measures used in the study—MCBS, Dirty Dozen, self-esteem scale, and self-efficacy scale—are utilized for the first time in the Romanian population? If I am wrong, I recommend supporting descriptions of all measures with citations of relevant papers demonstrating the psychometric soundness of the Romanian versions. Otherwise, I recommend conducting additional psychometric evaluations on the given sample. Although it would be impossible to fully test convergent and discriminant validity for all measures, at least exploratory or confirmatory factor analysis is possible to verify the factor validity. The authors reported values of Cronbach’s alpha that showed satisfactory to good internal consistency reliability for most psychological scales. However, high values of Cronbach’s alpha do not imply unidimensionality of scale (see Cortina, 1993; Green, Lissitz, & Mulaik, 1965). That is why factor analysis is required to gain proper support for the factorial structure. If the inclusion of additional analyses is at odds with space limitation, the authors can prepare a separate file with detailed information on the results of EFA/CFA and attach it as supplementary material.

Q17. The authors reported who participated in the translation of scales in the Romanian language but did not provide details on how translation and cultural adaptation was organized. Could the authors provide more details on the procedure of scale translation? Was forward translation followed by backward translation?

Q18. Lines 427–428. The Cronbach’s alpha coefficients indicated that the Dirty Dozen had satisfactory reliability.” I admit there is no consensus on what level of alpha to be considered satisfactory, but I find it hard to agree that the internal consistency of the psychopathy scale (α = .50) is satisfactory.

Results

Q19. Table 2. Given that the Table’s header refers to the variables as numbers (i.e., 1, 2, 3, etc.), it is advised to add numbers before the names of all measures in the column named “Variables” (i.e., change “Hurting People” to “1. Hurting People”, “Lying” to “2. Lying”, etc.).

Q20. Lines 452–453. Moderate positive associations were identified between psychopathy, narcissism, and Machiavellianism, which were consistent with prior study (Paulus, Willams, 2002).” I am not sure if commenting on such a finding is relevant given that there were no hypotheses about correlations among Dark Triad traits. At the same time, I do not consider it a crucial issue and prefer leaving it to the authors’ opinion as to whether to omit or preserve it.

Q21. Lines 454–489. I suppose there is no need to conduct both ANOVA and Tukey HSD in testing hypotheses 1–2. The reason is that both ANOVA and Tukey HSD keep the probability of Type I errors on the nominal level (α = .05). Thus, Tukey HSD is self-sufficient in the case of one-factor models. However, I prefer leaving it to the authors’ judgment.

Q22. Lines 454–489. In reporting results for Hypotheses 1–2, the authors provided effect sizes for ANOVA (eta-squared), but for post-hoc pairwise comparisons, the authors consistently refer to statistically significant results (p < .05) as “significant” without supplying their claims with effect size measures. Hence, I suggest accompanying descriptive statistics on the level of pairwise comparisons with values of effect sizes (e.g., Cohen’s d).

Q23. Lines 454–489. All post-hoc comparisons were reported as done with the Tukey HSD test. However, the design is unbalanced since the sample size across groups varies drastically (n = 15–85). In such circumstances, it is usually recommended to utilize the Tukey-Kramer test (Dunnett, 1980).

Q24. Lines 454–489. Hypotheses 1–3 predicted differences among groups for all subscales of MCBS, but only statistically significant results were discussed (e.g., differences between age groups in the hurting others are highlighted but similar results for lying and playing tricks are ignored). The latter creates reporting bias. Moreover, it may indirectly inflate publication bias in future meta-analyses. In sum, I strongly encourage the authors to describe both statistically significant and non-significant results.

Q25. Line 480.To test hypothesis 3, an independent samples t-test was conducted…” Was it a Student’s t-test or Welch’s t-test?

Q26. Lines 480–489. Hypothesis 3 posited that “violent offenders will show higher levels of malevolent creative ideation than non-violent offenders.” In other words, the difference was expected only in one particular direction (Mviolent > Mnon-violent). If so, I could not fully comprehend why the authors preferred a two-tailed test instead of a one-tailed test? The use of one-tailed tests would be more aligned with the prior hypothesis.

Q27. Lines 501–519. In paragraphs dedicated to the multiple regression, the authors presented some p-values as inequalities. Although p-values lower than .001 are indeed better to be written as inequalities (e.g., p < .001), all p-values higher than .001 should be written as exact numbers (e.g., p = 0.042). The rationale is twofold. First, p-values—when used in the logic of Fisher’s paradigm—quantify evidence against the null hypothesis, with lower values related to less consistency of empirical evidence with the null (Hubbard & Bayarri, 2003). Thus, reporting exact values of probability should be in priority. Second, as has been argued in many sources on the interpretation of p-values, p = .016 is not equivalent to p < .016 (see line 517; further details are available in Goodman, 2008).

Discussion

Q28. Lines 530–534. The first paragraph of the discussion recaps the purpose of the study but doesn’t mention one of the major spices of the paper: the investigation of malevolent creative ideation on the sample of culprits. Maybe such information can be also integrated into the first paragraph?

Q29. Lines 535–537. Our results support that those who commit in early ages crimes (16-20 years) have higher malevolent creativity ideation, on the hurting people subscale, supporting our first hypothesis.” Hypothesis 1 predicted differences between age groups not only for the subscale of hurting others but also for lying and playing tricks. Thus, it would be more accurate to tell that Hypothesis 1 has been partially supported.

Q30. Lines 535–537. Malevolent creativity, on the other hand, has been shown to have little or no relationship with emotional instability, but a stronger association with psychopathy and emotional coldness.” If it describes the results of previous studies, it would be better to back it up with a reference to the particular source.

Q31. Lines 546–547. As a result, it is not surprising that older rulebreakers are associated with malevolent creativity (due to emotional stability).” I haven’t grasped how this conclusion follows from the previous train of thought. Please could the authors elaborate on it?

Q32. Lines 601–602. As we hypothesized for lying, hurting people, and playing pranks, psychopathy showed no predictive power for malevolent creativity ideation.” This statement contradicts the hypothesis formulated in the literature review. Hypothesis 6 was stated as the following: “Psychopathy will significantly predict all forms of malevolent creative ideation.” (lines 319–320).

Q33. Lines 640–641. “…as Reiter-Palmon (180) notes out in his statement...” It would be more accurate to replace “his statement” with “her statement.”

Q34. Lines 633–650. I suggest discussing the low to moderate reliability of such subscales as playing tricks (MCBS) and psychopathy (Dirty Dozen) as a potential limitation of the study. Suboptimal reliability implies high measurement error which could be the reason why both scales haven’t shown any interesting patterns of association.

Reviewer 3 Report

The paper evaluates the interconnection between the Dark Triad traits and criminal behavior in order to predict malevolent creative ideation. The authors declare the possibility to use the Machiavellianism and self-efficacy traits as predictors of malevolent creative ideation in the form of lying. The reported study is of certain interest but I have several questions for the authors.

First, the penitentiaries are known to be places with a high concentration of people prone to malevolent behavior. This fact can significantly bias the reported study. Do authors have some "reference sample" from a less specific social group? For example, one can compare the personality traits of students and penitentiary residents. The authors note, that there are many similar studies for non-criminals, but barely compare their results with the mentioned ones.

My second concern follows from the first one. In such a closed and specific society as penitentiary residents, feedbacks are really important and should be taken into account unless all 130 inmates are in solitary confinements. The investigated social system is very complex and most of its feedbacks and connections may be hidden from the external observers.

Do some groups (gangs?) exist in the studied sub-society? If yes, how does it affect the results of the reported study?

The authors should be careful with claiming all inmates criminals. Eastern European penitentiary systems are known for sentence mistakes. Moreover, it is not obvious that all inmates possess "criminal minds". Some of them are just regular people which appeared in the wrong place at the wrong time. Many people are on their way to correction.

Fourth, the hypothesis "that a longer and more violent criminal career would be associated with higher levels of malevolent creativity" is strange from the beginning. For example, a low-organized sociopath can have a long and very violent "criminal career" possessing no creativity at all, both usual and malevolent. In many cases, the length of a criminal career depends more on law enforcement, than on the criminal's personality.

Does aggression necessarily require new ideas? I believe, non-creative aggression exists as well.

It is of great interest to consider female inmates as well. I believe, the gender influence can be significant here. Are your conclusions applicable to women's behavior?

The results are weakly supported by data due to the small sample. Using statistical methods on such a small and specific sample is questionable. Moreover, I barely understand how was the sincerity of respondents controlled in a case when they earned credits for participation in this study. Can their answers be a part of malevolent misleading creative behavior too?

Reviewer 4 Report

The article ‘Personality Traits as Predictors of Malevolent Creative Ideation in Offenders’ is interesting, well prepared and described. It moves, among others, the issues of associations of personality traits, including the dark triad, with malevolent creative ideation. The introduction is fully and thoroughly prepared. The characteristics of the study group were presented in a satisfactory way, and the research tools used were thoroughly described, as well as, which was valuable, the parameters that were taken into account when selecting the statistical tests. The results are presented clearly, the discussion is conducted in an interesting way. It is worth mentioning that the authors undertook the study of an interesting research group. The psychological functioning of offenders continues to require updated research.

Below are some minor suggestions for improving the manuscript:

1. In the material and methods part, it would be good to clearly present what statistical procedures were carried out in the study. Although their indication is dispersed in the text, a section devoted to a specific indication of the statistical tests that were used to assess the correlation, significance of differences and the assessment of predictions would be ordering.

2. In table 2, it would be worth to number the names of the variables so that it would be easier to read the correlations eg 1. Hurting People 2. Lying ...

3. I propose to fix minor writing errors - line 63 - 'asgood' shouldn't it be 'as good'? - line 75 'Goal type and the goal achievement is ...' in my opinion it should be 'Goal type and the goal achievement are the predictors' - please check the correct form.

Thank you for the opportunity to review the paper and congratulations to the authors for a good and interesting preparation of the article.

Round 2

Reviewer 2 Report

Response to revision

Even though the primary version of the manuscript was good, I am delighted to see how all adjustments made by the authors improved its quality even further. I sincerely thank the authors for their attention to detail and their hard-work attitude demonstrated during the revision process.

Almost all my comments were taken into consideration except for several ones. The authors briefly answered those comments in the following way: “We couldn’t operate this.” I am lost on how to interpret such replies. Nevertheless, I hypothesize that there could be at least three sources of confusion. First, I could fail to explain what I meant unambiguously. Second, the authors could be keen to make required adjustments but have difficulty in performing them. Lastly, the authors do not agree with my comments.

I suggest several ways of overcoming these sources of confusion. To mitigate the first one, I will reiterate and elaborate on my points below to make them more clear-cut. To mitigate the second one, I will provide additional details on how to perform requested changes in SPSS. With regards to the third one, I would like to make it clear that I respect the authors’ rights to present their study in the way they see as the most relevant and appropriate. There is no trouble if the authors disagree with any of the comments listed below. But it would benefit the process of peer-review if the authors could elaborate on their rebuttal by stating their perspectives. I do not insist on these adjustments and thus would like to leave it to the editor as to whether to require such changes in the manuscript or not. The discussed comments are provided below.

Q1. Lines 486–491. In reporting results for Hypotheses 1–2, the authors provided effect sizes for ANOVA (eta-squared), but for post-hoc pairwise comparisons, the authors consistently refer to statistically significant results as “significant” without supplying their claims with effect size measures. Hence, I suggest accompanying descriptive statistics on the level of pairwise comparisons with values of effect sizes (e.g., Cohen’s d). The potential benefit is that it would be easier for the authors to claim that the first hypothesis was supported for hurting others subscale based on effect size values rather than statistical significance (i.e., the mean difference between Group 1 and Group 2 was not statistically significant, but it was nonetheless of d = 0.56, corresponding to medium effect size). The authors used SPSS 22, which does not provide Cohen’s d by default. Thus, one can use online calculators to compute it. Here is one example of an online calculator from the University of Colorado: https://lbecker.uccs.edu/. To compute Cohen’s d, one has to enter either descriptive statistics for two groups or t value with degrees of freedom.

Q2. Lines 486–491. All post-hoc comparisons were reported as done with the Tukey HSD test. However, the design is unbalanced since the sample size across groups varies drastically (n = 15–85). In such circumstances, it is usually recommended to utilize the Tukey-Kramer test (Dunnett, 1980). The authors replied the following: “For wide variations of sample size, the Tukey-Kramer test may be preferred.” Two points. First, the variation from 15 to 85 participants is wide enough. Second, as reported on the official website of IBM, SPSS performs Tukey–Kramer test by default when a design is unbalanced and the Tukey HSD option is chosen (see https://www.ibm.com/support/pages/does-spss-offer-tukey-kramer-post-hoc-tests). All in all, I suggest changing the name of Tukey HSD to Tukey–Kramer test on page 10 in lines 486 and 500.

Q3. Lines 506–515. Hypothesis 3 posited that “violent offenders will show higher levels of malevolent creative ideation than non-violent offenders.” In other words, the difference was expected only in one particular direction (Mviolent > Mnon-violent). If so, I could not fully comprehend why the authors preferred a two-tailed test instead of a one-tailed test? The use of one-tailed tests would be more aligned with the prior hypothesis. If the problem lies in the technical domain, one can perform one-sided t-test here: short URL. At the same time, if the authors decide to use one-sided tests (Mviolent > Mnon-violent), they can also accompany it with other one-sided alternative (Mviolent < Mnon-violent) so that not to miss their finding in the opposite direction.

Q4. Lines 531–545. In paragraphs dedicated to multiple regression, the authors presented some p-values as inequalities. Although p-values lower than .001 are indeed better to be written as inequalities (e.g., p < .001), all p-values higher than .001 should be written as exact numbers (e.g., p = 0.042). The rationale is twofold. First, p-values—when used in the logic of Fisher’s paradigm—quantify evidence against the null hypothesis, with lower values related to less consistency of empirical evidence with the null (Hubbard & Bayarri, 2003). Thus, reporting exact values of probability should be in priority. Second, as has been argued in many sources on the interpretation of p-values, p = .016 is not equivalent to p < .016 (see line 543; further details are available in Goodman, 2008). To make it clear, I refer to two concrete p-values: (1) p < .05 (for self-efficacy beta weight reported in line 533) and (2) p < .16 (for regression model reported in line 543). Please consider recomputing the regression models to extract these p-values with precision to the third decimals and replacing inequalities with these exact p-values.

Minor comments

After reading the revised manuscript, I identified only a small portion of minor comments.

Q5. Lines 46–57. Other definitions of creativity could be conceived of as a product that stands out for its value and originality (8).” Three points. First, it seems misleading to compare “other definitions”—emphasizing originality and value—with that of Sternberg and Lubart (1996) whom themselves stated that creativity is an ability to produce something novel and appropriate (Sternberg & Lubart, 1996, p. 677). Second, in my opinion, it would be more accurate to make a distinction between definitions of creativity and creativity conceptualizations. Sternberg and Lubart (1996) and “other definitions” refer to how creativity is defined, while Rhode’s (1961) and Cropley and Cropley’s (2010) works refer to identifying facets of creativity. Lastly, pay attention that you referred to Cropley and Cropley (2010) as simply “Cropley” (compare in-text citation in line 48 with reference list in lines 731–733).

Q6. Lines 451–452. Other authors as well, as Florin et all. (154) translated into Romanian, total DT .85, Narcissism .86, Psychopaty .64 and Machiavellianism .81, in a sample of students.” Thanks a lot for reporting additional evidence on internal consistency reliability! However, don’t you mind making it clear that all numbers reported in the sentence are Cronbach’s alphas? It will help to avoid confusion.

Q7. Lines 453–457. The paragraph on self-esteem is better to be separated from the paragraph on Dirty Dozen.

Q8. Lines 504–505. The subscales for lying and playing tricks have showed no correlation with the number of convictions.” Instead of correlation, it is worth shifting the focus to comparing averages. Otherwise, one may wonder why the authors used ANOVA in one case and correlation coefficients in the other.

Q9. Lines 506–515. Please consider explicitly mentioning somewhere in the paragraph that the mean difference was found in the opposite direction from predicted. Otherwise, a potential reader may not fully grasp why Hypothesis 3 was not confirmed if mean differences were statistically significant.

Q10. Line 542. The third multiple regression model statistically failed to significantly...” The word “statistically” is undue.

Q11. Line 553. Only Machiavellianism significantly predicted malevolent ideation for the sub-scale hurting people, but not for the playing tricks and lying subscales...” I think it would be better to remove the phrase “but not for the playing tricks and lying subscales” because later in the sentence, the authors write that Machiavellianism predicted malevolent ideation for the lying subscale.

Q12. Lines 566–571. Discussing the study of Jia et al. (2020), the authors flavored their narration with a bunch of statistical values labeled as “p =…” What is p here? If it is a correlation, I advise changing p to r. Please correct also the first author’s surname from “Jin” to “Jia” in the text (see line 566).

Q13. Line 573. those who commit in early 572 ages crimes (16-20 years) have higher malevolent creativity ideation…” Higher than who? Please specify the proper reference group(s).

Author Response

Response to revision

Even though the primary version of the manuscript was good, I am delighted to see how all adjustments made by the authors improved its quality even further. I sincerely thank the authors for their attention to detail and their hard-work attitude demonstrated during the revision process.

Almost all my comments were taken into consideration except for several ones. The authors briefly answered those comments in the following way: “We couldn’t operate this.” I am lost on how to interpret such replies. Nevertheless, I hypothesize that there could be at least three sources of confusion. First, I could fail to explain what I meant unambiguously. Second, the authors could be keen to make required adjustments but have difficulty in performing them. Lastly, the authors do not agree with my comments.

We may have given an incomplete answer in certain cases to the initial round of comments because we were unable to react to the comments you submitted, due to a number of determinable factors.

We made now efforts to fix all of you’re comments, so please feel free to tell us if there is anything to change in our paper.

Your comments enabled us to make significant improvements, so we appreciate your support and advice as well as your constructive viewpoint. As a result, our paper has improved.

Once more, we want to express our gratitude for your time and effort.

I suggest several ways of overcoming these sources of confusion. To mitigate the first one, I will reiterate and elaborate on my points below to make them more clear-cut. To mitigate the second one, I will provide additional details on how to perform requested changes in SPSS. With regards to the third one, I would like to make it clear that I respect the authors’ rights to present their study in the way they see as the most relevant and appropriate. There is no trouble if the authors disagree with any of the comments listed below. But it would benefit the process of peer-review if the authors could elaborate on their rebuttal by stating their perspectives. I do not insist on these adjustments and thus would like to leave it to the editor as to whether to require such changes in the manuscript or not. The discussed comments are provided below.

Q1. Lines 486–491. In reporting results for Hypotheses 1–2, the authors provided effect sizes for ANOVA (eta-squared), but for post-hoc pairwise comparisons, the authors consistently refer to statistically significant results as “significant” without supplying their claims with effect size measures. Hence, I suggest accompanying descriptive statistics on the level of pairwise comparisons with values of effect sizes (e.g., Cohen’s d). The potential benefit is that it would be easier for the authors to claim that the first hypothesis was supported for hurting others subscale based on effect size values rather than statistical significance (i.e., the mean difference between Group 1 and Group 2 was not statistically significant, but it was nonetheless of d = 0.56, corresponding to medium effect size). The authors used SPSS 22, which does not provide Cohen’s d by default. Thus, one can use online calculators to compute it. Here is one example of an online calculator from the University of Colorado: https://lbecker.uccs.edu/. To compute Cohen’s d, one has to enter either descriptive statistics for two groups or t value with degrees of freedom.

We have added Cohen’s d for all differences, as suggested. Although this is not traditionally expected in reporting results, we agree that it is informative for the reader, both for significant and non-significant results.

Q2. Lines 486–491. All post-hoc comparisons were reported as done with the Tukey HSD test. However, the design is unbalanced since the sample size across groups varies drastically (n = 15–85). In such circumstances, it is usually recommended to utilize the Tukey-Kramer test (Dunnett, 1980). The authors replied the following: “For wide variations of sample size, the Tukey-Kramer test may be preferred.” Two points. First, the variation from 15 to 85 participants is wide enough. Second, as reported on the official website of IBM, SPSS performs Tukey–Kramer test by default when a design is unbalanced and the Tukey HSD option is chosen (see https://www.ibm.com/support/pages/does-spss-offer-tukey-kramer-post-hoc-tests). All in all, I suggest changing the name of Tukey HSD to Tukey–Kramer test on page 10 in lines 486 and 500.

Changed the name of Tukey HSD to Tukey–Kramer test

Q3. Lines 506–515. Hypothesis 3 posited that “violent offenders will show higher levels of malevolent creative ideation than non-violent offenders.” In other words, the difference was expected only in one particular direction (Mviolent > Mnon-violent). If so, I could not fully comprehend why the authors preferred a two-tailed test instead of a one-tailed test? The use of one-tailed tests would be more aligned with the prior hypothesis. If the problem lies in the technical domain, one can perform one-sided t-test here: short URL. At the same time, if the authors decide to use one-sided tests (Mviolent > Mnon-violent), they can also accompany it with other one-sided alternative (Mviolent < Mnon-violent) so that not to miss their finding in the opposite direction.

The simplest approach here recognises that the significance level for the symmetrical two-tailed t-test (assuming significance of 0.05) allocates half of the significance value (i.e., 0.025) to each tail. Therefore, the one-tailed significance – as recommended by IBM itself – is simply half the two-tailed figure. We have reported these now as one-tailed. The result is not altered – H3 (specifically) as written, is not supported. It is, of course, interesting that the result was the reverse of what we expected.

Q4. Lines 531–545. In paragraphs dedicated to multiple regression, the authors presented some p-

values as inequalities. Although p-values lower than .001 are indeed better to be written as inequalities (e.g., p < .001), all p-values higher than .001 should be written as exact numbers (e.g., p = 0.042). The rationale is twofold. First, p-values—when used in the logic of Fisher’s paradigm—quantify evidence against the null hypothesis, with lower values related to less consistency of empirical evidence with the null (Hubbard & Bayarri, 2003). Thus, reporting exact values of probability should be in priority. Second, as has been argued in many sources on the interpretation of p-values, p = .016 is not equivalent to p < .016 (see line 543; further details are available in Goodman, 2008). To make it clear, I refer to two concrete p-values: (1) p < .05 (for self-efficacy beta weight reported in line 533) and (2) p < .16 (for regression model reported in line 543). Please consider recomputing the regression models to extract these p-values with precision to the third decimals and replacing inequalities with these exact p-values.

Good point. The first (p < .05) has been corrected to p = .043 The second case was a typo (< instead of =) and this has been corrected.

Minor comments

After reading the revised manuscript, I identified only a small portion of minor comments.

Q5. Lines 46–57. Other definitions of creativity could be conceived of as a product that stands out for its value and originality (8).” Three points. First, it seems misleading to compare “other definitions”—emphasizing originality and value—with that of Sternberg and Lubart (1996) whom themselves stated that creativity is an ability to produce something novel and appropriate (Sternberg & Lubart, 1996, p. 677). Second, in my opinion, it would be more accurate to make a distinction between definitions of creativity and creativity conceptualizations. Sternberg and Lubart (1996) and “other definitions” refer to how creativity is defined, while Rhode’s (1961) and Cropley and Cropley’s (2010) works refer to identifying facets of creativity. Lastly, pay attention that you referred to Cropley and Cropley (2010) as simply “Cropley” (compare in-text citation in line 48 with reference list in lines 731–733).

The intent here is to highlight the fact that there is a wide range of ways that creativity is defined and conceptualized. We changed “Other definitions” to “Other descriptions” to avoid suggesting that we are only talking about “definitions”. Cropley “& Cropley” inserted.

Q6. Lines 451–452. Other authors as well, as Florin et all. (154) translated into Romanian, total DT .85, Narcissism .86, Psychopaty .64 and Machiavellianism .81, in a sample of students.” Thanks a lot for reporting additional evidence on internal consistency reliability! However, don’t you mind making it clear that all numbers reported in the sentence are Cronbach’s alphas? It will help to avoid confusion.

Corrected, as you suggested, “Other authors as well, as Florin et all. (154)  translated into Romanian, total DT (Cronbach’s α =.85), Narcissism (Cronbach’s α =.86), Psychopathy (Cronbach’s α =.64) and Machiavellianism (Cronbach’s α =.81), in a sample of 168 students”

Q7. Lines 453–457. The paragraph on self-esteem is better to be separated from the paragraph on Dirty Dozen.

The two paragraphs have been separated from each other.

Q8. Lines 504–505. The subscales for lying and playing tricks have showed no correlation with the number of convictions.” Instead of correlation, it is worth shifting the focus to comparing averages. Otherwise, one may wonder why the authors used ANOVA in one case and correlation coefficients in the other.

The subscales for lying and playing tricks have showed no correlation with the number of convictions.” The sentences was deleted.

Q9. Lines 506–515. Please consider explicitly mentioning somewhere in the paragraph that the mean difference was found in the opposite direction from predicted. Otherwise, a potential reader may not fully grasp why Hypothesis 3 was not confirmed if mean differences were statistically significant.

Now that we clarified this in the results section (with the one-tailed results) we have also reiterated this result here. We state this in the section already (“Contrary to our third hypothesis….” However we added a couple of words to further emphasise this.

Q10. Line 542. The third multiple regression model statistically failed to significantly...” The word “statistically” is undue. 

Deleted the word statistically

Q11. Line 553. Only Machiavellianism significantly predicted malevolent ideation for the sub-scale hurting people, but not for the playing tricks and lying subscales...” I think it would be better to remove the phrase “but not for the playing tricks and lying subscales” because later in the sentence, the authors write that Machiavellianism predicted malevolent ideation for the lying subscale.

We removed “but not for the playing tricks and lying subscales”, as you suggested.

Q12. Lines 566–571. Discussing the study of Jia et al. (2020), the authors flavored their narration with a bunch of statistical values labeled as “p =…” What is p here? If it is a correlation, I advise changing p to r. Please correct also the first author’s surname from “Jin” to “Jia” in the text (see line 566).

Changed the surname Jin to Jia, and yes, you observed right that there are correlational data provided.

Q13. Line 573. those who commit in early 572 ages crimes (16-20 years) have higher malevolent creativity ideation…” Higher than who? Please specify the proper reference group(s).

Completed the sentence to make it fully understanding.
